# Comparison of bone mineral density of runners with inactive males: A cross-sectional 4HAIE study

Miroslav Krajcigr[1]*, Petr Kutáč[1], Steriani Elavsky[1], Daniel Jandačka[1], Matthew Zimmermann[2]

1 Department of Human Movement Science, University of Ostrava, Ostrava, Czech Republic, 2 School of Human Sciences, The University of Western Australia, Crawley, WA, Australia

* miroslav.krajcigr@osu.cz

## Abstract

The purpose of the study was to determine whether running is associated with greater bone mineral density (BMD) by comparing the BMD of regularly active male runners (AR) with inactive nonrunner male controls (INC). This cross-sectional study recruited 327 male AR and 212 male INC (aged 18–65) via a stratified recruitment strategy. BMD of the whole body (WB) and partial segments (spine, lumbar spine (LS), leg, hip, femoral neck (FN), and arm for each side) were measured by dual-energy x-ray absorptiometry (DXA) and lower leg dominance (dominant-D/nondominant-ND) was established by functional testing. An ANCOVA was used to compare AR and INC. The AR had greater BMD for all segments of the lower limb (p<0.05), but similar BMD for all segments of the upper limb (p>0.05) compared with INC. Based on the pairwise comparison of age groups, AR had greater BMD of the ND leg in every age group compared with INC (p<0.05). AR had grater BMD of the D leg in every age group except for (26–35 and 56–65) compare with INC (p<0.05). In the youngest age group (18–25), AR had greater BMD in every measured part of lower extremities (legs, hips, femoral necks) compared with INC (p<0.05). In the 46–55 age group AR had greater BMD than INC (p < 0.05) only in the WB, D Leg, D neck, and ND leg. In the 56–65 age group AR had greater BMD than INC (p<0.05) only in the ND leg. Overall, AR had greater BMD compared with INC in all examined sites except for the upper limbs, supporting the notion that running may positively affect bone parameters. However, the benefits differ in the skeletal sites specifically, as the legs had the highest BMD difference between AR and INC. Moreover, the increase in BMD from running decreased with age.

## Introduction

Osteoporosis is a major public health problem as more than 8.9 million fractures occur annually, resulting in an osteoporotic fracture every 3 seconds [1]. A 10% loss of bone mass in the hip can result in 2.5 times higher risk of hip fractures, and similarly, a 10%loss of bone mass in the vertebrae can double the risk of vertebral fracture [2]. In addition, low calcaneal bone

**Data Availability Statement:** Available data from Program 4 HAIE can be found at https://www. 4haie.cz/en/data-2/ (accessed on 12 August 2021) and results and publications from Program 4 HAIE

can be found at https://www.4haie.cz/publikace/ (accessed on 1 July 2022).

**Funding:** This research was funded by Program 4 HAIE—Healthy Aging in Industrial Environment (grant number: CZ.02.1.01/0.0/0.0/16_019/ 0000798). This article has been produced with the financial support of the European Union under the project LERCO (CZ.10.03.01/00/22_003/0000003) via the Operational Program Just Transition. The baseline data refers to the project funded by the Czech Ministry of Education, Youth and Sports, the project 4HAIE "Healthy Aging in the Industrial Environment - Program 4" (CZ.02.1.01/0.0/0.0/ 16_019/0000798) within its sustainability period.

**Competing interests:** The authors have declared that no competing interests exist.

mineral density (BMD) was found as an independent risk factor for hip fractures in a prospective study of 9517 females aged $\geq$65 [3]. High costs are also associated with osteoporosis, as the number of fractures is projected to grow annually and surpass 3 million by 2025 in the United States, costing an estimated $25 billion [4].

It is believed that the reconstruction of bone tissue takes place throughout the life cycle with bone mass increasing within the first two decades, before plateauing by the beginning of the fourth decade [5, 6]. Once bone mass has plateaued, it slowly decreases throughout the remaining years for both males and females [7–9]. Warming et al. [9] reported that the rate of bone loss after reaching peak BMD (~30 years of age) in males was ~0.22–0.33%/year at all sites except at the femoral neck (0.42%/year). However, in females a more specific approach is required when assessing bone loss [10] due to the possible influence of age of menarche, menstrual irregularities, and menopause [9, 11].

Epidemiological studies of families and twin suggest that 60%-80% of the diversity of BMD values in adulthood is due to heredity [12, 13]. The remaining 20%-40% can be affected by exogenous factors such as nutrition, smoking, and physical activity (PA). No single factor can be neglected to achieve the maximum BMD [12, 14]. Bones have many vital functions throughout the life cycle, and each year, can dynamically construct 10% of the skeleton [15]. Once the remodeling starts, it takes up to 4 months until the new bone structure is completely created [16]. Therefore, bone health is of great importance, and large intervention and epidemiological studies have shown a positive relationship between bone health and PA [17].

In particular, running can positively influence BMD due to the axial load on the skeleton during the running motion [18]. Running is considered an impact, weight-bearing exercise, which causes an increase in BMD through the impact of ground reaction forces and muscle strain generated during running [19–21]. Additionally, repeated bounces and impacts occur during running, and the body is exposed to impact loads. The shock wave which occurs in running, is initiated by contact of the runners' feet with the ground and is transmitted by the skeletal system to the head of the runner [19, 20]. Therefore, this could influence the BMD of all segments of the body.

These findings are supported by several studies that have shown higher BMD in runners compared to controls [22–24] or suggested that running may help maintain BMD [25]. However, multiple studies [26–28] have also found that the type of activity benefits specific sites of the skeleton. Higher BMD values were shown at the skeletal sites experiencing a greater increase in load from running (e.g., calcaneus, lower limbs) compared to the sites experiencing less load (e.g., spine) [26, 29–38]. However, some studies did not find a positive association between running and BMD [29, 39]. Mitchell et al. [39] found no differences in BMD values between middle-aged long-term endurance runners and controls. Additionally, in a study comparing the BMD of sprinters and endurance runners, no differences were identified for BMD in the hip or spine between endurance runners and controls. However, sprinters had greater hip and spine BMD compared with endurance runners and controls [29]. Previous research has also observed an adverse association between running and bone mass [40, 41]. Düz et al. [40] identified that ultramarathon runners had significantly lower BMD values than active males and the control group. Similarly, high-level endurance runners had lower BMD in the lumbar spine than non-athletes [41]. Therefore, there is a certain amount of PA that can allow for a beneficial increase in BMD to be attained.

Based on current literature, the association between running and bone health is still unclear. Most of the studies mentioned previously focused only on adolescent and collegiate runners [11, 23, 28], older men and the elderly [25, 29, 40], on elite and high-level runners [22, 26, 27, 41], or had a small sample size [39]. Thus, a large cohort of recreational active runners (AR) and inactive nonrunner controls (INC) of various ages would help to address the gap in

the current understanding of this specific population. Additionally, based on the specification of the studies in women [10], we decided to focus this study only on men. The aim of this study was to compare BMD of male regular AR with that of INC to determine whether running is associated with greater BMD. We hypothesized that AR would have greater BMD in the sites experiencing a greater increase in load from running compared with INC.

## Material and methods

### Subjects

A total of 446 AR and 263 INC aged 18–65 years completed the 4HAIE study (CZ.02.1.01/0.0/ 0.0/16_019/0000798). Of these, many had missing data for the variables required in this study (see in statistical analysis) and were not included in the analysis. This left 327 AR and 212 INC with complete data available for analysis. The study participants were recruited via a stratified recruitment strategy from 1/3/2019 to 30/8/2021, with the assistance of a professional marketing and social science research company. A full description of the participants and inclusion and exclusion criteria is shown previously published 4HAIE research [42–44].

Based on the screening survey, subjects were divided into two groups, AR and INC. AR needed to meet the WHO public PA guidelines [45] (150 min/week, moderate or 75 min/vigorous PA, or a combination of both) and be running a minimum of 10 km per week for at least one year prior to the study. Furthermore, based on the PA Survey, the AR took part in 3.28 ± 1.38 running sessions per week, and in each session, they ran 8.87 ± 3.20 km in 52.24 ± 19.36 minutes. In total, this is on average 29.69 ± 18.87 km/week and 175.00 ± 114.26 min/week with an average pace of 6.06 ± 1.72 min/km. INC were those that did not meet WHO public PA guidelines. All participants were nonsmokers and signed an informed consent—see more in Ethics and dissemination. All participants had no acute disease, no acute health conditions or chronic diseases (within the six weeks prior) that would prevent them from PA (pain, injury, surgery). Participants were excluded if they had undergone radiological examination in the seven days prior to measurement or if they had an artificial pacemaker, radioactive, surgical, or any other device/implant or insulin pump. The authors did not have access to information that could identify individual participants during or after data collection.

### Procedures

All the data were derived as part of the 4HAIE project. A more detailed description of the entire measurement protocol can be found in previously published 4HAIE research [42–44], and an overview of all assessment tools are available at https://www.4haie.cz/en/data-2/.

Those who were interested in participating completed screening online and by telephone. Eligible participants were assigned to a group (AR/INC) based on the screening survey, and the laboratory assessment was scheduled. Informed consent was obtained in person upon arrival to the laboratory. An additional questionnaire was then completed on a computer in the laboratory and included questions about PA and running history. The questionnaire on PA and running history also served as verification for grouping based on the screening survey.

Somatic measurements included body height, body mass, and body composition. All measurements were taken in the morning on the second day. The participants were measured in sports clothing (shorts and T-shirt) and barefoot. Firstly, body height and mass were measured using an InBody 370 stadiometer (Biospace, South Korea). Body composition was then measured using dual-energy x-ray absorptiometry (DXA) method. BMD was measured using the Hologic Horizon A bone densitometer (Hologic Discovery A, Waltham, MA). Whole-body scans were used for body composition analysis, including the segmental analysis. The results

for the upper and lower limbs were derived from segmental analysis of whole-body scan. In addition, specific site scans were used for the hip (right, left including femoral neck) and lumbar spine (L1-L4). The body position during these measurements and the scans of the individual areas measured can be found in the Hologic manual [46]. The recommended position with the hips flexed at 90 degrees was used during the lumbar spine (L1-L4) scan, and the dual hip feet position was used for the hip scans. The measured parameters were body fat, fat free mass, bone mineral content, and BMD.

To establish the dominant lower limb, the functional "kick ball" test was used [47]. Participants were asked to kick an imitation of a soccer ball with moderate intensity and maximal accuracy to the imitation of a soccer goal, which was 1 m wide and 10 m from the participants. Three trials were conducted, and the leg used for most trials was identified as the dominant limb.

### Ethics and dissemination

The study was approved by the ethics committee of the University of Ostrava (protocol code OU-87674/90-2018 and date of approval 29 November 2018) and was in compliance with the 1964 Helsinki Declaration and its later amendments or comparable ethical standards. Before providing written, informed consent, a detailed participant information sheet was provided to each participant.

### Statistical analysis

Statistical analysis was performed using Excel 2016 (Microsoft Corporation) and IBM SPSS Statistics (IBM, Armonk, NY, USA) for Windows. An independent sample $t$-test was used to compare basic characteristics of AR and INC. The effect size (ES) was calculated based on the Cohen´s $d$ and was estimated as follows $d > 0.2$ small, $d > 0.5$ medium, $d > 0.8$ large and $d > 1.2$ very large [48]. The ES was established as significant if $d > 0.5$. To compare BMD of AR with BMD of INC and eliminate confounding variables, ANCOVA with Bonferroni correction was used. To evaluate the difference in BMD across different age groups and activity status, ANCOVA with Bonferroni correction was performed. According to the age groups of HAIE, the participants were assigned into age (18–25, 26–35, 36–45, 46–55, 56–65) and activity status (AR, INC) groups. Moreover, a pairwise comparison of AR and INC within the same age group was conducted. To further investigate factors associated with BMD, stepwise linear regression was used for all participants. At each step, variables were chosen based on $p$-values, and a $p$-value threshold of 0.05 was used to set a limit on the total number of variables included in the final model. The results of stepwise linear regression are shown as standardized regression coefficient (β), 95% confidence interval (CI), and $p$ values for the best model, including $R^2$. Statistical significance for analysis was established at $p < 0.05$.

The dependent variables were BMD of whole-body (WB), spine, lumbar spine (LS), dominant leg, dominant hip, dominant femoral neck, nondominant leg, nondominant hip BMD, nondominant femoral neck, left arm, and right arm. For ANCOVA analyses, the controlling variables (age, mass, height, BMI, fat mass and lean mass) were determined based on previous research [11, 22, 23, 26–29, 49]. Moreover, based on the previous research [11, 22, 23, 26–29, 39, 49] the possible predictors of BMD for stepwise linear regression were established (activity status, age, mass, height, BMI, fat mass, and lean mass).

### Results

Table 1 shows the basic characteristics of AR and INC, the independent sample $t$-test, and the effect size of Cohen´s $d$. Data are shown as mean ± standard deviation (SD) along with the

**Table 1. Comparison of age, somatic characteristics, and running distance and time of AR vs INC.**

| Variable | AR M ± SD | INC M ± SD | MD | p | 95% CI of the Difference | Effect size (d) |
|---|---|---|---|---|---|---|
| Age (years) | 37.76 ± 8.59 | 40.13 ± 13.60 | -2.36 | 0.027 | -4.45 to -0.27 | 0.20 |
| Mass (kg) | 81.08 ± 9.77 | 88.54 ± 13.94 | -7.46 | 0.000 | -9.47 to -5.45 | 0.65 |
| Height (cm) | 180.54 ± 6.49 | 180.69 ± 6.79 | -0.15 | 0.801 | -1.29 to 1.00 | 0.02 |
| BMI (kg/m$^2$) | 24.27 ± 2.52 | 26.50 ± 4.24 | -2.23 | 0.000 | -2.80 to -1.66 | 0.68 |
| Fat mass (kg) | 19.06 ± 4.69 | 26.65 ± 7.81 | -7.59 | 0.000 | -8.65 to -6.53 | 1.24 |
| Lean mass (kg) | 62.02 ± 6.70 | 61.89 ± 7.59 | -0.13 | 0.839 | -1.10 to 1.35 | 0.02 |
| Running distance (km/week) | 29.69 ± 18.87 | 0.54 ± 1.91 | +29.15 | 0.000 | 26.60 to 31.71 | 1.98 |
| Running time (min/week) | 175.00 ± 114.27 | 3.73 ± 13.41 | +171.23 | 0.000 | 155.74 to 186.72 | 1.92 |

AR–active runners, INC–inactive nonrunner controls, M–mean, SD–standard deviation, MD–mean difference, p–p value, CI–confidence interval, d–Cohen´s d, BMI–body mass index.

mean difference (*MD*). There was no significant difference between AR and INC in height, and lean mass (p > 0.05). The two groups were statistically and practically different in other characteristics (age, mass, fat mass, and BMI), with AR having lower values of mass, fat mass and BMI (p < 0.05).

## Comparison of all AR and INC

The results of ANCOVA analyses in Table 2 showed a significant difference at each measured BMD site when comparing BMD of AR and INC of WB, spine, LS, dominant leg, dominant hip, dominant femoral neck, nondominant leg, nondominant hip, nondominant femoral neck, left arm and right arm (p < 0.05). The smallest difference was in spine and lumbar spine (p < 0.05). The only two measured sites which did not show a significant difference were left arm and right arm (p > 0.05).

## BMD across the age and activity group

The distribution into groups according to age and activity status was as follows: AR 18–25 n = 60, AR 26–35 n = 85, AR 36–45 n = 103, AR 46–55 n = 64, AR 56–65 n = 15, INC 18–25

**Table 2. BMD comparison of AR vs INC.**

| Variable | AR M ± SD | INC M ± SD | F | p |
|---|---|---|---|---|
| WB BMD (g/cm$^2$) | 1.16 ± 0.08 | 1.11 ± 0.09 | +7.721 | 0.006 |
| Spine BMD (g/cm$^2$) | 1.06 ± 0.13 | 1.03 ± 0.14 | +5.757 | 0.017 |
| LS BMD (g/cm$^2$) | 1.05 ± 0.13 | 0.99 ± 0.15 | +4.142 | 0.042 |
| Dominant leg BMD (g/cm$^2$) | 1.26 ± 0.09 | 1.19 ± 0.10 | +22.800 | 0.000 |
| Dominant hip BMD (g/cm$^2$) | 1.08 ± 0.13 | 1.03 ± 0.13 | +11.838 | 0.001 |
| Dominant femoral neck BMD (g/cm$^2$) | 0.96 ± 0.14 | 0.89 ± 0.14 | +14.741 | 0.000 |
| Nondominant leg BMD (g/cm$^2$) | 1.26 ± 0.09 | 1.18 ± 0.10 | +28.039 | 0.000 |
| Nondominant hip BMD (g/cm$^2$) | 1.09 ± 0.13 | 1.03 ± 0.13 | +8.263 | 0.004 |
| Nondominant femoral neck BMD (g/cm$^2$) | 0.95 ± 0.15 | 0.89 ± 0.14 | +10.693 | 0.001 |
| Left arm BMD (g/cm2) | 0.84 ± 0.06 | 0.81 ± 0.52 | +0.912 | 0.340 |
| Right arm BMD (g/cm2) | 0.86 ± 0.06 | 0.82 ± 0.06 | +0.786 | 0.376 |

AR–active runners, INC–inactive nonrunner controls, M–mean, SD–standard deviation, F–F value, p–p value, WB–whole body, LS–lumbar spine, BMD–bone mineral density.

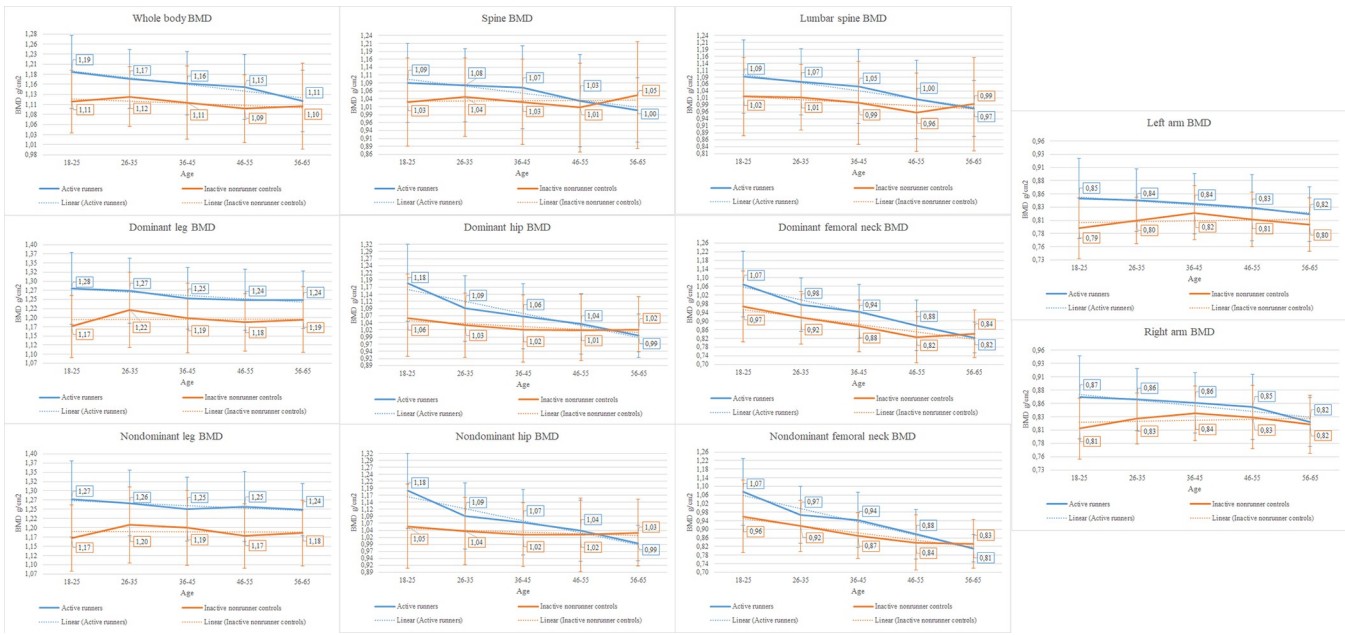

**Fig 1. Graphs of BMD means of different age groups and measured sites.**

n = 42, INC 26–35 n = 44, INC 36–45 n = 52, INC 46–55 n = 42, INC 56–65 n = 32. A graphical representation of the BMD values of a specific age and activity group is presented in Fig 1. Fig 1 shows BMD means of different age and activity status groups of every measured site. The difference between AR and INC was different across all age categories. A more detailed comparison of each individual age group can be found in the ANCOVA analysis.

An ANCOVA analysis revealed that there was not a significant interaction between the effects of age and activity status on the BMD of every measured site, controlling for mass, height, BMI, fat mass and lean mass.

The analysis showed that after controlling for mass, height, BMI, fat mass and lean mass, activity status had a significant effect on the BMD of the WB (F = 6.533 p = 0.011), spine (F = 3.986, p = 0.046), dominant leg (F = 23.158, p = 0.000), dominant hip (F = 14.180, p < 0.001), dominant femoral neck (F = 14.688, p < 0.001), nondominant leg (F = 29.167, p = 0.000), nondominant hip (F = 9.377, p = 0.002), and nondominant femoral neck (F = 10.390, p = 0.001). Conversely, activity status did not have a significant effect on the BMD of the LS (F = 2.742, p = 0.098), left arm (F = 1.124, p = 0.289), and right arm (F = 0.313 p = 0.576). The analysis also showed that age group did have a significant effect on BMD of WB (F = 2.848, p = 0.023, spine (F = 3.488, p = 0.008), LS (F = 4.753, p = 0.001), dominant hip (F = 21.808, p = 0.000), dominant femoral neck (F = 41.573, p < 0.001), right nondominant hip (F = 18.292, p < 0.001), and right nondominant femoral neck (F = 39.308, p < 0.001). Simple main effects analysis showed that age group did not have a significant effect on BMD of dominant leg (F = 1.436, p = 0.221), nondominant leg (F = 0.563, p = 0.690), left arm (F = 0.417, p = 0.797), and right arm (F = 0.739, p = 0.566).

An ANCOVA pairwise comparison of AR and INC in the same age group is presented in Table 3. In every age group, AR had greater BMD in the nondominant leg compared with INC (p < 0.05). In every age group except for (26–35 and 56–65), AR had greater BMD in the dominant leg (p < 0.05) The nondominant leg was the only measured site where was a significant difference in the oldest (56–65) age groups. In the youngest age group (18–25), AR had greater

**Table 3. ANCOVA pairwise comparisons of BMD of different measured site for AR vs INC in the same age group.**

| Age group | WB Diff (p) | Spine Diff (p) | LS Diff (p) | D leg Diff (p) | D hip Diff (p) | D f neck Diff (p) | ND leg Diff (p) | ND hip Diff (p) | ND f neck Diff (p) | L arm Diff (p) | R arm Diff (p) |
|---|---|---|---|---|---|---|---|---|---|---|---|
| **18–25** | +0.026 (0.099) | +0.029 (0.266) | +0.026 (0.339) | +0.065 (0.000) | +0.081 (0.000) | +0.061 (0.011) | +0.068 (0.000) | +0.084 (0.000) | +0.066 (0.008) | +0.014 (0.191) | +0.017 (0.104) |
| **26–35** | +0.015 (0.286) | +0.040 (0.103) | +0.034 (0.183) | +0.029 (0.064) | +0.055 (0.010) | +0.054 (0.017) | +0.038 (0.021) | +0.048 (0.026) | +0.037 (0.107) | +0.009 (0.350) | +0.005 (0.469) |
| **36–45** | +0.028 (0.030) | +0.051 (0.021) | +0.046 (0.045) | +0.038 (0.007) | +0.046 (0.015) | +0.064 (0.002) | +0.036 (0.015) | +0.042 (0.031) | +0.065 (0.002) | -0.004 (0.632) | -0.002 (0.859) |
| **46–55** | +0.036 (0.019) | +0.037 (0.149) | +0.036 (0.174) | +0.052 (0.002) | +0.036 (0.106) | +0.068 (0.004) | +0.069 (0.000) | +0.025 (0.260) | +0.047 (0.050) | +0.006 (0.563) | +0.005 (0.657) |
| **56–65** | +0.005 (0.824) | -0.011 (0.779) | -0.016 (0.693) | +0.042 (0.100) | +0.019 (0.575) | +0.010 (0.775) | +0.055 (0.042) | -0.002 (0.950) | +0.005 (0.895) | +0.006 (0.689) | -0.011 (0.496) |

Diff.–difference, p–p value, WB–whole body, LS–lumbar spine, D–dominant, ND–nondominant, L–left, R–right, f–femoral.

BMD in every measured part of lower extremities (legs, hips, femoral necks) compared with INC ($p < 0.05$). In the 36–45 age group, AR had greater BMD in every measured site except for the left and right arm compared with INC ($p < 0.05$). The upper limbs were the only site which did not showed any significant age group difference.

## Possible predictors of BMD

The results of stepwise linear regression are shown in Tables 4 and 5. Starting with seven variables that might theoretically based on the previous research [11, 22, 23, 26–29, 39, 49] be good predictors of BMD, a stepwise linear regression was able to reduce them to 3 or 4 with dependence on the site. The activity status was included in every model except for the upper limbs,

**Table 4. Results for whole and upper body final model of the stepwise linear regression presented as standardised regression coefficient (β).**

| Variable | WB BMD ($R^2 = 0.316$) β | 95% CI | p | Spine BMD ($R^2 = 0.198$) β | 95% CI | p | LS BMD ($R^2 = 0.188$) β | 95% CI | p |
|---|---|---|---|---|---|---|---|---|---|
| **Activity status** | -0.131 | -0.039 to -0.008 | 0.004 | -0.116 | -0.054 to -0.011 | 0.003 | -0.098 | -0.055 to -0.001 | 0.045 |
| **Age (years)** | -0.082 | -0.001 to 0.000 | 0.000 | -0.106 | -0.002 to 0.000 | 0.007 | -0.148 | -0.003 to -0.001 | 0.000 |
| **Mass (kg)** | -0.497 | 0.000 to 0.000 | 0.000 | | | | -0.257 | 0.000 to 0.000 | 0.008 |
| **Height (cm)** | | | | | | | | | |
| **BMI (kg/m²)** | | | | | | | | | |
| **Fat mass (kg)** | | | | | | | | | |
| **Lean mass (kg)** | 0.842 | 0.000 to 0.000 | 0.000 | 0.421 | 0.000 to 0.000 | 0.000 | 0.562 | 0.000 to 0.000 | 0.000 |

| Variable | Left arm BMD ($R^2 = 0.368$) β | 95% CI | p | Right arm BMD ($R^2 = 0.382$) β | 95% CI | p |
|---|---|---|---|---|---|---|
| **Activity status** | | | | | | |
| **Age (years)** | | | | | | |
| **Mass (kg)** | -0.744 | 0.000 to 0.000 | 0.000 | | | |
| **Height (cm)** | -0.155 | -0.002 to -0.001 | 0.000 | -0.107 | -0.002 to 0.000 | 0.009 |
| **BMI (kg/m²)** | | | | | | |
| **Fat mass (kg)** | | | | -0.431 | 0.000 to 0.000 | 0.000 |
| **Lean mass (kg)** | 1.182 | 0.000 to 0.000 | 0.000 | 0.748 | 0.000 to 0.000 | 0.000 |

$R^2$ –coefficient of determination, β –standardised regression coefficient, CI–confidence interval, p–p value, WB–whole body, LS–lumbar spine, BMD–bone mineral density, BMI–body mass index.

**Table 5. Results for lower body final model of the stepwise linear regression presented as standardised regression coefficient (β).**

| Variable | Dominant leg BMD ($R^2 = 0.328$) | | | Dominant hip BMD ($R^2 = 0.318$) | | | Dominant femoral neck BMD ($R^2 = 0.382$) | | |
|---|---|---|---|---|---|---|---|---|---|
| | β | 95% CI | p | β | 95% CI | p | β | 95% CI | p |
| Activity status | -0.199 | -0.056 to -0.022 | 0.000 | -0.172 | -0.065 to -0.027 | 0.000 | -0.190 | -0.076 to -0.036 | 0.000 |
| Age (years) | | | | -0.317 | -0.004 to -0.003 | 0.000 | -0.443 | -0.006 to -0.004 | 0.000 |
| Mass (kg) | -0.434 | 0.000 to 0.000 | 0.000 | | | | | | |
| Height (cm) | | | | -0.196 | -0.006 to -0.002 | 0.000 | -0.102 | -0.004 to 0.000 | 0.013 |
| BMI (kg/m²) | | | | | | | | | |
| Fat mass (kg) | | | | | | | | | |
| Lean mass (kg) | 0.796 | 0.000 to 0.000 | 0.000 | 0.542 | 0.000 to 0.000 | 0.000 | 0.469 | 0.000 to 0.000 | 0.000 |
| Variable | Nondominant leg BMD ($R^2 = 0.321$) | | | Nondominant hip BMD ($R^2 = 0.300$) | | | Nondominant femoral neck BMD ($R^2 = 0.382$) | | |
| | β | 95% CI | p | β | 95% CI | p | β | 95% CI | p |
| Activity status | -0.233 | -0.065 to -0.030 | 0.000 | -0.160 | -0.062 to -0.024 | 0.000 | -0.180 | -0.075 to—0.034 | 0.000 |
| Age (years) | | | | -0.293 | -0.004 to -0.002 | 0.000 | -0.443 | -0.006 to -0.005 | 0.000 |
| Mass (kg) | | | | | | | | | |
| Height (cm) | -0.127 | -0.004 to 0.000 | 0.048 | -0.195 | -0.006 to -0.002 | 0.000 | -0.122 | -0.005 to -0.001 | 0.003 |
| BMI (kg/m²) | | | | | | | | | |
| Fat mass (kg) | | | | | | | | | |
| Lean mass (kg) | 0.691 | 0.000 to 0.000 | 0.000 | 0.541 | 0.000 to 0.000 | 0.000 | 0.484 | 0.000 to 0.000 | 0.000 |

$R^2$ –coefficient of determination, β –standardised regression coefficient, CI–confidence interval, p–p value, WB–whole body, LS–lumbar spine, BMD–bone mineral density, BMI–body mass index.

and running was shown as a protective factor. For the whole body BMD, the analysis reduced variables to 4, which were: lean mass, mass, activity status and, age. The spine BMD analysis included lean mass, activity status, and age. For the LS BMD analysis, the stepwise linear regression reduced the variables to 4, which were: lean mass, mass, age, and activity status. The dominant leg BMD model included lean mass, mass, and activity status. In regard to the dominant hip BMD model, the variables were reduced to 4, which were: lean mass, height, age, and activity status. For the dominant femoral neck BMD analysis, the stepwise linear regression model was able to reduce variables to 4 as follows: lean mass, age, activity status, and height. The nondominant leg BMD model included lean mass, activity status, and height. The nondominant hip BMD analysis variables were reduced to four: lean mass, age, height, and activity status. For the nondominant femoral neck BMD, the analysis reduced variables to 4, which were: lean mass, age, activity status, and height. The left and right arm were the only two measured sites which did not include activity status in the final model. For the left and arm BMD analysis, the stepwise linear regression reduced the variables to 3, which were: lean mass, mass, and height and for the right arm BMD analysis, the stepwise linear regression reduced the variables to 3, which were: lean mass, fat mass and height.

## Discussion

This cross-sectional study examined the BMD of 327 AR and 212 INC aged 18 to 65 with similar height and lean mass. The AR had lower values of mass, fat mass, and BMI. When comparing the BMD of every measured site, the AR had significantly higher BMD values at each measured BMD site except for the left and right arm.

Based on the ANCOVA analysis of all AR and INC in the current study, the findings that AR have greater BMD compared with INC are consistent with that of previous research. Hind

et al. [22] measured the BMD of 31 male endurance runners aged 18–35 years and found that at runners had greater age, height and weight adjusted BMD of the left total proximal femur than controls (p<0.05). Similarly, Saers et al. [24] found increased trabecular BMD in 15 male distance runners aged 23.4 ± 3.3 years compared with sedentary controls. In addition, Infantino et al. [23] observed greater BMD at total hip and whole body in 21 male collegiate athletes aged 18–23 years compared with 22 male controls (PA and exercising energy expenditure < 500 kcal/day) matched for height, BMI, and age. Nonetheless, these studies [22–24] did not study men throughout the lifespan, instead focusing on younger adults. In comparison, the current study included men across the life span (from 18 to 65 years old) and showed that after controlling for age, mass, height, BMI, fat mass and lean mass AR had higher BMD than INC at all measured sites except for left and right arm. The upper limbs were used as controlling BMD site, which should not be directly affected by running. However, the difference tended to be smaller in the spine and lumbar spine site compared to the lower extremities and its sub sites (hip and femoral neck). There was potential site dependence in BMD as possibly more loaded sites (i.e., legs, calcaneus, hip, and femoral neck) had greater values compared to the sites experiencing less load (i.e., lumbar spine) [26, 29–38]. A possible explanation for lower lumbar spine values compared to the other sites may be because the LS is a predominantly trabecular bone, which may be less influenced by lower body impact loading and local muscle action [19, 20, 50].

Furthermore, as seen in Fig 1, the difference is not the same for all sub-age groups. In further investigation of age and status group as two fixed factors ANCOVA analysis showed no significant interaction between the effects of age and activity status on BMD of every measured site. However, further investigation showed a significant effect of activity status alone on the BMD of every measured site except for the lumbar spine, left arm and right arm. Additionally, age group alone had a significant effect on the BMD of the WB, spine, LS, dominant hip, dominant femoral neck, nondominant hip, and nondominant femoral neck. When using a pairwise comparison for AR and INC in the same age group, the middle age group (36–45) was significantly different in every measured site, except for the left and right arm. In the 18–25 age group, there was a significant difference in every measured site of the lower extremities (legs, hips, and femoral necks). Those findings are also congruent with previous studies on the same age group (18–35) [22–24], showing the higher BMD values of runners in the same age range. In the older age groups, there was no difference in the majority of measured sites. This may be due to bone maturation [5, 6] as the three youngest groups were more prone to bone stimulation by PA (running), and the influence of PA might diminish with age as the bone remodelling cycle differs in later age.

In the older age group, the influence of running on BMD may only occur in sites that receive additional loading through running itself. This study showed that AR in every age group had significantly greater BMD of the nondominant and dominant leg (with the exception of 26–35 years age group for the dominant leg) compared with INC. Site dependence has been established before in multiple studies [26–28]. In the vast majority, the difference was between the more loaded sites (lower limbs and their subparts) and the sites experiencing less load (spine and lumbar spine). When comparing the AR and INC, we found greater values at each loaded site, but the difference was smaller at the spine and lumbar spine compared to other sites. Additionally, Fig 1 shows that the dominant and nondominant leg had a lower decline of BMD between the age groups compared to the BMD of the hip and femoral neck BMD. The dominant and the nondominant leg were also the only two measured sites that differed in the ANCOVA pairwise comparison of every age group, except for the 26–35 age group of dominant leg. Therefore, the results indicate that the higher BMD values must be in the lower part of the lower extremities and that the site dependence is enhanced with age. It

suggests the confirmation of the load site dependence [26–28] and shows that the shock wave caused by running motion [19, 20] plateaus in the transition from the lower extremities of the runners to the head. However, other possible factors, such as: participation in other sports with high impacts or weight-lifting training may also explain the increases in BMD seen, although to establish the influence of these variables was not the objective of this study. To mitigate this limitation, we compared the BMD of the left and right arm of AR with INC, which were similar between the two groups.

Compared with the findings from studies that showed no difference between middle-aged long-term endurance runners and controls [39], and between master endurance runners and controls [29], the presented study included a broader sample than master athletes or middle-aged men alone. Mitchell et al. [39] found no difference between middle-aged (46–55 years old) long-term endurance runners and controls. In comparison, in the presented study, AR aged between 46–55 years old had greater BMD of the WB, left leg, right leg, and right femoral neck compared with INC. The differences with the findings of Mitchell et al. [39] may be due to the different study groups. The AR in this study ran 29.17 ± 19.50 km/week, whereas the endurance runners in the study of Mitchell et al. [39] ran 82.6 (± 27.9) km/week. Additionally, the study of Piasecki et al. [29] consisted of master athletes, which tend to have a high running mileage that could be associated with a possible negative effect on the BMD in runners [36]. The higher mileage might also explain the incongruent results as lower BMD has previously been seen in middle-aged ultramarathon male runners compared to sedentary controls [40]. Furthermore, lower BMD in the lumbar spine was observed in high-level endurance runners compared to non-athletes [41]. Therefore, the running mileage or the specification of participants (age and the size of the sample) may explain the differences in BMD seen.

Some studies [27, 35, 37, 50–54] have tried to address the association between BMD and running mileage. Burrows et al. [51] showed that greater running mileage was negatively associated with lumbar spine and femoral neck BMD. Hetland et al. [37], who compared elite runners and controls, found those who ran more than 100 km/week, on average, 19% lower lumbar bone mineral content. MacDougall et al. [53] found no difference in BMD of the spine and trunk in groups with different running mileage. However, in the lower limb of a control sedentary group, BMD increased with increasing running miles/week, peaking at 24–32 km/week. Following this, BMD decreased with increasing miles/week, with 64–86 km/week and 97–120 km/week indicating the lowest BMD. Moreover, Barrack and colleagues [54] identified that running >48 km/week was one of the strongest risk factors in predicting low BMD of lumbar spine, however it was not for WB. Other research has indicated that the possible mileage beneficial threshold for bone health may be somewhere between 90–100 km/week [35, 37, 52].

However, to our knowledge, no unanimous beneficial threshold for BMD and mileage has been established as heterogenous samples were used and other factors such as genetics, nutrition, metabolic factors could play a significant role as well [27, 34, 50, 51]. Based on analysis of our data (S1 File) the mileage cut-off is likely greater than 105 km/week, which supports previous research that speculated it could be above the 100 km/week [35, 37, 52]. However, research has observed decreases in BMD in those running >32 km/week [53]. Further large sample studies with specific data on the amount of running are required before a more accurate threshold can be achieved.

## Strength and limitation

The strength of this paper is the large sample size of a cohort of AR and INC through a wide life span (from 18 to 65 years old). Another strength is that participants remained in standardized conditions for 16 hours prior to the measurements. However, this is a cross-sectional

study of Caucasian men, and the generalization to other groups must be taken with caution. Despite the large overall sample size, there were unequal sample sizes within age and activity status groups. Moreover, the study lacked information on other important determinants of BMD, such as childhood history of PA, nutrition (such as diet, food frequency etc.), or genetics [6, 11, 55]. Furthermore, other PA that may influence BMD such as strength training, aerobic training, and individual or team sport participation (including level of expertise) were not controlled in the analyses and could have impacted the results seen [22, 26, 32, 33, 56–59]. On the other hand, the analysis was controlled for variables (age, mass, height, BMI, fat mass and lean mass) found to be associated with BMD in previous research [11, 22, 23, 26–29, 49]. Additionally, based on the study design, the causal relationships cannot be postulated. Furthermore, PA /running were assessed by self-report, which has been known to be subject bias. To minimize this issue, we triangulated information from different questionnaires to verify running status. This study did not focus on the possible significant covariance as the biomechanical variables and running mileage that could influence the BMD.

## Conclusion

This cross-sectional study showed that AR had greater BMD in all examined sites than inactive nonrunner controls except for the upper limbs, supporting the notion that running positively affects bone parameters. However, the benefits differ in the skeleton sites specifically, as the legs had the highest BMD difference between AR and INC. The results suggested that the shock wave generated in the running motion plateaus in the transition from the lower extremities to the head of the runner. However, the benefits appear to dimmish with age.

## Supporting information

**S1 File. BMD mileage threshold.**
(ZIP)

## Author Contributions

**Conceptualization:** Miroslav Krajcigr, Petr Kutáč, Daniel Jandačka.

**Data curation:** Petr Kutáč, Steriani Elavsky, Daniel Jandačka.

**Funding acquisition:** Daniel Jandačka.

**Investigation:** Petr Kutáč, Steriani Elavsky, Daniel Jandačka.

**Methodology:** Miroslav Krajcigr, Daniel Jandačka.

**Project administration:** Daniel Jandačka.

**Resources:** Daniel Jandačka.

**Validation:** Miroslav Krajcigr, Petr Kutáč, Steriani Elavsky.

**Visualization:** Miroslav Krajcigr.

**Writing – original draft:** Miroslav Krajcigr, Petr Kutáč, Steriani Elavsky.

**Writing – review & editing:** Miroslav Krajcigr, Petr Kutáč, Steriani Elavsky, Daniel Jandačka, Matthew Zimmermann.

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
