## [Decision Letter · Decision Letter 0]

13 Sep 2023

PONE-D-23-16917Comparison of bone mineral density of runners with inactive males: A Cross-Sectional 4HAIE StudyPLOS ONE

Dear Dr. Krajcigr,

Thank you for submitting your manuscript to PLOS ONE. After careful consideration, we feel that it has merit but does not fully meet PLOS ONE’s publication criteria as it currently stands. Therefore, we invite you to submit a revised version of the manuscript that addresses the points raised during the review process.

We look forward to receiving your revised manuscript.

Kind regards,

Enock Madalitso Chisati, PhD

Academic Editor

PLOS ONE

4. We note that Figures 1 and 2 in your submission contain copyrighted images. All PLOS content is published under the Creative Commons Attribution License (CC BY 4.0), which means that the manuscript, images, and Supporting Information files will be freely available online, and any third party is permitted to access, download, copy, distribute, and use these materials in any way, even commercially, with proper attribution. For more information, see our copyright guidelines: http://journals.plos.org/plosone/s/licenses-and-copyright.

1. You may seek permission from the original copyright holder of Figures 1 and 2 to publish the content specifically under the CC BY 4.0 license.

Additional Editor Comments:

In the methods section of the abstract, provide a summary of study site, how participants were recruited. Also indicate that these were male runners. Mention the partial body segments where BMD was measured.

In the results section of the abstract, provide figures in brackets to back the statistical significance of the comparisons. Provide a summary of results of the pairwise comparison for AR and INC in the same age group. Figures are important to the reader to verify the differences.

Reviewers' comments:

Reviewer's Responses to Questions

**Comments to the Author**

1. Is the manuscript technically sound, and do the data support the conclusions?

Reviewer #1: Yes

2. Has the statistical analysis been performed appropriately and rigorously? 

Reviewer #1: Yes

3. Have the authors made all data underlying the findings in their manuscript fully available?

Reviewer #1: Yes

4. Is the manuscript presented in an intelligible fashion and written in standard English?

Reviewer #1: No

5. Review Comments to the Author

Reviewer #1: Thank you for giving me the opportunity to review this paper. It is interesting and much work has gone into its preparation, however, there are multiple grammatical errors and clarity problems (for example first sentence on page 4). There are words missing (e.g., page 4, starting with "Furthermore...") and incomplete sentences (e.g., in last paragraph page 20, starting with "compared with..."

In my opinion there are several issues that need to be addressed before the manuscript can get considered for publication:

1. the conclusion that 'running is good for BMD' has been discussed before, so the novelty is not there. However, is it possible that too much running is NOT good for the BMD of the spine? The paper does not discuss how much running is good (and the amount of running was not measured). I believe that the literature has been trying to establish that there is a limit after which running becomes unfavorable to spine BMD.

2. A major weakness of this study was the criterion that was chosen to be considered 'active' vs. non-active. 'Running for 6 weeks (or less) prior to data collection' is not a good discriminator between "active" and non-active participants. This short amount of time might have consequences to the cardiovascular system, but I doubt that it could affect BMD

3. BMD of the right leg vs. left leg was compared, it should have been 'dominant' vs. 'non-dominant' leg

4. page 21: the authors discuss the 'unloading' of the spine, but at the same time explain how the loading or 'shock' wave travels through the spine. I don't believe that the spine is 'unloaded', but it is indeed loaded during running. There are many references in the literature that support this notion.

5. "Mitchell" is spelled with 2 'l'

6. The inclusion of hundreds of participants is definitely a strength of this study, but it gets weakened by lack of variables that were controlled, e.g. running speed, running time, days and miles/km per week, etc. and the amount of body areas that were compared. I believe that one or two focused area (for example the lumbar spine, since this is the only area that has been the point of discussion in terms of BMD) would have made this study stronger.

6. PLOS authors have the option to publish the peer review history of their article (what does this mean?). If published, this will include your full peer review and any attached files.

Reviewer #1: No

---

## [Author Response · Author response to Decision Letter 0]

22 Nov 2023

Answer: Template was used for addressing the manuscript requirements. Additionally, the naming and style requirements have been read and followed.

Answer: Thank You for pointing out. We wanted to address this issue in revision and re-submission, however in the resubmitting process we were not able to change the statement in the ‘Financial Disclosure’ to ‘This research was funded by Program 4 HAIE—Healthy Aging in Industrial Environment (grant number: CZ.02.1.01/0.0/0.0/16_019/0000798). The funders had no role in study design, data collection and analysis, decision to publish, or preparation of the manuscript.’

Answer: We have edited the text in the method section and deleted the statement from the other section. 

4. We note that Figures 1 and 2 in your submission contain copyrighted images. All PLOS content is published under the Creative Commons Attribution License (CC BY 4.0), which means that the manuscript, images, and Supporting Information files will be freely available online, and any third party is permitted to access, download, copy, distribute, and use these materials in any way, even commercially, with proper attribution. For more information, see our copyright guidelines: http://journals.plos.org/plosone/s/licenses-and-copyright.

1. You may seek permission from the original copyright holder of Figures 1 and 2 to publish the content specifically under the CC BY 4.0 license.

Answer:

Figure 1 and 2 were taken from the Hologic Manual (2013) and modified to show the necessary information more clearly and concisely. A citation and reference have been added to the text.

However, if this is not a sufficient clarification and explanation, please let us know, we will address this issue differently. 

Additional comments:

1. Thank you for confirming where you have obtained the images in Figure 1 and 2, and for including the corresponding citation in your manuscript.

Should your paper be accepted, all images will be published under PLOS’ Creative Commons Attribution (CC BY) 4.0 license, which means that they will be freely available online, and any third party is permitted to access, download, copy, distribute, and use these materials in any way, even commercially, with proper attribution. For more information, see our Figure guidelines: http://www.plosone.org/static/figureGuidelines#policies

With regards to the previously published and copyrighted figure in your submission, Figures 1 and 2, we require you to either present written permission from the copyright holder to publish this figure, or remove the figures from your submission. We have noted that the figure was published by the Hologic, Inc. and, therefore, ask that you please obtain permission from this company.

1. To seek permission from the Hologic, Inc. to publish Figures 1 and 2 under the specific Creative Commons Attribution License (CCAL), CC BY 4.0, please contact them with the following text:

“I request permission for the open-access journal PLOS ONE to publish XXX under the Creative Commons Attribution License (CCAL) CC BY 4.0 (http://creativecommons.org/licenses/by/4.0/). Please be aware that this license allows unrestricted use and distribution, even commercially, by third parties. Please reply and provide explicit written permission to publish XXX under a CC BY license.”

Please upload the granted permission to the manuscript as an other file. In the figure caption of the copyrighted figure, please include the following text: “Republished from [ref] under a CC BY license, with permission from [name of publisher], original copyright [original copyright year].”

Please note that RightsLink permission forms often impose use restrictions that are incompatible with our CC BY 4.0 license, and we are therefore unable to accept these permissions. For this reason, we strongly recommend contacting copyright holders with the PLOS ONE Request for Permission form.

2. If you are unable to obtain permission from the journal, please either A) remove the figure or B) supply a replacement figure that complies with the CC BY 4.0 license. Please check copyright information on all replacement figures and update the figure caption with source information. If applicable, please specify in the figure caption text when a figure is similar but not identical to the original image used in the study, and is therefore for illustrative purposes only.

Additional answer:

Thank you for clarifying the possible solutions to this problem. After careful consideration and taking into account many circumstances, we have decided to remove Figures 1 and 2 from the text and leave only the reference to the relevant manual. Based on these changes, we have also adjusted the numbering of the retained Figures.

Additional Editor Comments:

In the methods section of the abstract, provide a summary of study site, how participants were recruited. Also indicate that these were male runners. Mention the partial body segments where BMD was measured.

In the results section of the abstract, provide figures in brackets to back the statistical significance of the comparisons. Provide a summary of results of the pairwise comparison for AR and INC in the same age group. Figures are important to the reader to verify the differences.

Answer: 

We edited the text. Information about the recruitment, sex and measured body segments were added.

In the result section we provided figures in brackets. Additionally, we provided the pairwise comparison.

Reviewers' comments:

Reviewer's Responses to Questions

Comments to the Author

1. Is the manuscript technically sound, and do the data support the conclusions?

Reviewer #1: Yes

2. Has the statistical analysis been performed appropriately and rigorously?

Reviewer #1: Yes

3. Have the authors made all data underlying the findings in their manuscript fully available?

Reviewer #1: Yes

4. Is the manuscript presented in an intelligible fashion and written in standard English?

Reviewer #1: No

Answer:

We used the academic English language tool Writefull. Moreover, a native English speaker from the kinanthropology academic field has revised the manuscript.

5. Review Comments to the Author

Reviewer #1: Thank you for giving me the opportunity to review this paper. It is interesting and much work has gone into its preparation, however, there are multiple grammatical errors and clarity problems (for example first sentence on page 4). There are words missing (e.g., page 4, starting with "Furthermore...") and incomplete sentences (e.g., in last paragraph page 20, starting with "compared with..."

Answer: Thank You for the kind words and moreover thank You for addressing the issues of this manuscript. To address grammatical errors and clarity problems we revised the whole text. Additionally, we used the academic English language tool Writefull. Moreover, a native English speaker from the kinanthropology academic field has revised the manuscript.

In my opinion there are several issues that need to be addressed before the manuscript can get considered for publication:

1. the conclusion that 'running is good for BMD' has been discussed before, so the novelty is not there. However, is it possible that too much running is NOT good for the BMD of the spine? The paper does not discuss how much running is good (and the amount of running was not measured). I believe that the literature has been trying to establish that there is a limit after which running becomes unfavorable to spine BMD.

Answer:

Thank You for this point.

We are aware of the fact, that some studies indicated that too much running may not be beneficial for bone health. E.g., Burrows et al. (2003) showed that greater running mileage was negatively associated with lumbar spine and femoral neck BMD. Additionally, weekly running distance was inversely associated lumbar spine BMD in the study of Hind et al. (2006). The concern for lumbar spine as a site prone to low BMD was in some other studies (Barrack et al., 2008; Bilanin et al., 1989; Fredericson et al., 2007; Goodpaster et al., 1996; Hetland et al., 1993; MacDougall et al., 1992; MacKelvie, 2000; Tam et al., 2018; Winters et al., 1996)

Greater BMD values for runners in more loaded sites (i.e., legs, calcaneus hip, Wards triangle, trochanter, and femoral neck) has been shown previously (Aloia et al., 1978; Brahm et al., 1997; Fredericson et al., 2007; Heinonen et al., 1993; Kohrt et al., 2004; Piasecki et al., 2018; Tam et al., 2018; Wolman et al., 1991). Likewise lower values for lumbar spine have been seen previously (Bilanin et al., 1989; Brahm et al., 1997; Fredericson et al., 2007; Goodpaster et al., 1996; Hetland et al., 1993; Tam et al., 2018). However, one study has shown greater BMD for lumbar spine (Lane et al., 1998).

Hind and colleagues (Hind et al., 2006) aimed to address the possible explanation for the difference in lumbar spine values compared to other sites. The reason may be due the lumbar spine as predominantly trabecular bone which may be less influenced by lower body impact loading and local muscle action.

Regarding the beneficial running mileage threshold:

To our knowledge, no unanimous threshold for mileage has been established as heterogenous samples were used and other factors such as genetics, nutrition, metabolic factors etc. could play a significant role as well (Burrows et al., 2003; Hind et al., 2006; Kemmler et al., 2006; Tam et al., 2018).

Some studies indicated that the threshold for proximal femur could exist, and it could be somewhere between 90-100 km/week (Bilanin et al., 1989; Hetland et al., 1993; MacKelvie, 2000). 

Regarding the lumbar spine, some studies found lower vertebral BMD in young adult runners whose training over ≈ 92 km/ week in compare with controls (Bilanin et al., 1989; Hetland et al., 1993). 

From those studies that were trying to focus on the mileage:

Hetland et al. (1993), whose compared elite runners and controls, found out that the difference increased with the weakly distance. Those who ran more than 100 km/week had the lumbar bone mineral content, on the average, 19% lower. Moreover, Lumbar BMC was negatively correlated with the weekly distance run (r = -0.37; P < 0.0001), with a difference of 19 +/- 5% (mean +/- SEM). A similar relation was also found for all measurement sites.

MacDougal et al. (1992) studied controls and runners in 5 mileage groups. The groups were as follows: controls, 5-10 miles/week, 15-20 miles/week, 25-30 miles/week, 40-55 miles/week, 60-75 miles/week.

They found no statistically significant difference in BMD of spine and trunk. However, BMD of lower legs was increasing from controls and 5-10 miles/week group to 15-20 miles/week group, where the values were the greatest. After this peak, decrease was showed thorough the 25-30 miles/week, 40-55 miles/week, and 60-75 miles/week groups.

Kemmler et al. (2006) observed a low, but not signiﬁcant association between training volume (km/week/year) and trabecular BMD of the femoral neck. This association disappeared when adjusting for age, BMI, and body fat in the group of highly trained male runners. They postulated that: “The eﬀect of long distance running per se on bone parameters is not deleterious.”

Barrack and colleagues (2017) identified risk factors for low BMD: running >30 miles/week was one of the strongest risk factors predicting low BMD of lumbar spine, however it was not for WB.

(Hind et al., 2006) presented in their manuscript that there were moderate negative correlations between weekly running distance and LS BMD (r2 = 0.267; 0.189; P < 0.001).

We were also considering adding this aim and information to the manuscript. However, we elected not to include it as the purpose of this study is different. Adequate information has been used and added to the introduction and discussion section.

Based on the aforementioned studies, we might presume that the threshold could be somewhere about 90-100 km/week. But as it is written abov

---

## [Decision Letter · Decision Letter 1]

28 May 2024

PONE-D-23-16917R1Comparison of bone mineral density of runners with inactive males: A Cross-Sectional 4HAIE StudyPLOS ONE

Dear Dr. Krajcigr,

Thank you for submitting your manuscript to PLOS ONE. After careful consideration, we feel that it has merit but does not fully meet PLOS ONE’s publication criteria as it currently stands. Therefore, we invite you to submit a revised version of the manuscript that addresses the points raised during the review process.

We look forward to receiving your revised manuscript.

Kind regards,

Enock Madalitso Chisati, PhD

Academic Editor

PLOS ONE

Reviewers' comments:

Reviewer's Responses to Questions

**Comments to the Author**

1. If the authors have adequately addressed your comments raised in a previous round of review and you feel that this manuscript is now acceptable for publication, you may indicate that here to bypass the “Comments to the Author” section, enter your conflict of interest statement in the “Confidential to Editor” section, and submit your "Accept" recommendation.

Reviewer #2: (No Response)

2. Is the manuscript technically sound, and do the data support the conclusions?

Reviewer #2: Partly

3. Has the statistical analysis been performed appropriately and rigorously? 

Reviewer #2: Yes

4. Have the authors made all data underlying the findings in their manuscript fully available?

Reviewer #2: Yes

5. Is the manuscript presented in an intelligible fashion and written in standard English?

Reviewer #2: Yes

6. Review Comments to the Author

Reviewer #2: Although the authors achieve the established objective, there are a set of variables that remain unexplained. Food frequency or other variables (strength training, professional activity) that may have implications for BMD stand out.

The fact that we are dealing with a "large" sample cannot be ignored, on the other hand there are many variables that remain uncontrolled.

The limitations should be clear that they were not controlled and that they could have implications on the results.

The tables need to be improved; they are difficult to read, particularly the 4.

7. PLOS authors have the option to publish the peer review history of their article (what does this mean?). If published, this will include your full peer review and any attached files.

Reviewer #2: No

---

## [Author Response · Author response to Decision Letter 1]

17 Jun 2024

1. If the authors have adequately addressed your comments raised in a previous round of review and you feel that this manuscript is now acceptable for publication, you may indicate that here to bypass the “Comments to the Author” section, enter your conflict of interest statement in the “Confidential to Editor” section, and submit your "Accept" recommendation.

Reviewer #2: (No Response)

2. Is the manuscript technically sound, and do the data support the conclusions?

Reviewer #2: Partly

3. Has the statistical analysis been performed appropriately and rigorously? 

Reviewer #2: Yes

4. Have the authors made all data underlying the findings in their manuscript fully available?

Reviewer #2: Yes

5. Is the manuscript presented in an intelligible fashion and written in standard English?

Reviewer #2: Yes

6. Review Comments to the Author

Reviewer #2: Although the authors achieve the established objective, there are a set of variables that remain unexplained. Food frequency or other variables (strength training, professional activity) that may have implications for BMD stand out.

The fact that we are dealing with a "large" sample cannot be ignored, on the other hand there are many variables that remain uncontrolled.

The limitations should be clear that they were not controlled and that they could have implications on the results.

Thank you for your comment and for highlighting additional variables that were not clearly stated. We have edited the text in the ‘Strengths and limitations’ section to include these variables. Unfortunately, we were not able to use these variables in the analyses because the data for this paper were taken from the 4HAIE project. The design of the 4HAIE project was not focused specifically in this direction and the variables you have specified were not evaluated in the 4HAIE study. On the other hand, the study accounted for other variables that could be used from the 4HAIE project and have been previously shown to be associated with BMD such as age, mass, height, BMI, fat mass and lean mass [11,22,23,26–29,49]. Additionally, because it was not possible to control for all variables affecting (potentially affecting) BMD, control segments (upper limbs) were used.

Specifically:

In the Strength and limitations section we have added additional information to the sentence that refers to nutrition to further specify nutrition. 

“Moreover, the study lacked information on other important determinants of BMD, such as childhood history of PA, nutrition (such as diet, food frequency etc.), or genetics.”

We have added a sentence to the limitations about variables that were not controlled and about possible bias in the results based on the absence of those variables. 

“Furthermore, other PA that may influence BMD such as strength training, aerobic training, and individual or team sport participation (including level of expertise) were not controlled in the analyses and could have impacted the results seen [22,26,32,33,56–59].”

We have also added information about the variables that were controlled to the Strength and limitations section so that the information about what was and was not controlled is as accurate as possible.

“On the other hand, the analysis was controlled for variables (age, mass, height, BMI, fat mass and lean mass) found to be associated with BMD in previous research [11,22,23,26–29,49].”

The tables need to be improved; they are difficult to read, particularly the 4.

Thank you for this comment. For all tables, we have adjusted the centred text alignment for the result values and their names to make them easier to read. The captions under all tables have been reviewed and adjusted to ensure they are as clear and comprehensive as possible, in conjunction with the labels within the tables themselves. Moreover, we focused on the most problematic tables separately.

Regarding Table 1, the numbers of subjects have been removed, as these are provided in the methodology section. Additionally, a more detailed description of the values in the table has been added (M ± SD, “d” added to Effect size).

Regarding Table 2, the numbers of subjects have been removed, as these are provided in the methodology section. Additionally, a more detailed description of the values in the table has been added.

Regarding Table 3: we have changed the orientation of the page to make the table more readable and to enable increasing the size of the font. We have added information indicating that the segments represent "differences and p values” in the table to enhance clarity when reading the table.

Regarding Table 4: we have divided this table into two tables. Specifically, into upper body and lower body. We further separated it from the text. Moreover, we did some additional formatting in the tables (spaces, borders, …).

---

## [Decision Letter · Decision Letter 2]

24 Jun 2024

Comparison of bone mineral density of runners with inactive males: A Cross-Sectional 4HAIE Study

PONE-D-23-16917R2

Dear Dr. Krajcigr,

We’re pleased to inform you that your manuscript has been judged scientifically suitable for publication and will be formally accepted for publication once it meets all outstanding technical requirements.

Kind regards,

Enock Madalitso Chisati, PhD

Academic Editor

PLOS ONE

Additional Editor Comments (optional):

Reviewers' comments:

Reviewer's Responses to Questions

**Comments to the Author**

1. If the authors have adequately addressed your comments raised in a previous round of review and you feel that this manuscript is now acceptable for publication, you may indicate that here to bypass the “Comments to the Author” section, enter your conflict of interest statement in the “Confidential to Editor” section, and submit your "Accept" recommendation.

Reviewer #2: (No Response)

2. Is the manuscript technically sound, and do the data support the conclusions?

Reviewer #2: Yes

3. Has the statistical analysis been performed appropriately and rigorously? 

Reviewer #2: Yes

4. Have the authors made all data underlying the findings in their manuscript fully available?

Reviewer #2: Yes

5. Is the manuscript presented in an intelligible fashion and written in standard English?

Reviewer #2: Yes

6. Review Comments to the Author

Reviewer #2: The concerns regarding confound variables have been addressed with a more complete "Strengths and Limitations" section as well as adequate literature support. The tables clarity has been improved. To note what appears to be an incomplete sentence in line 46 and the misspelling of "discussion" on the headline.

7. PLOS authors have the option to publish the peer review history of their article (what does this mean?). If published, this will include your full peer review and any attached files.

Reviewer #2: No

---

## [Editor Report · Acceptance letter]

3 Jul 2024

PONE-D-23-16917R2 

PLOS ONE

Dear Dr. Krajcigr, 

I'm pleased to inform you that your manuscript has been deemed suitable for publication in PLOS ONE. Congratulations! Your manuscript is now being handed over to our production team.

Kind regards, 

on behalf of

Dr. Enock Madalitso Chisati 

Academic Editor

PLOS ONE